# PAIR DIFFUSION: A COMPREHENSIVE MULTIMODAL OBJECT-LEVEL IMAGE EDITOR

## ABSTRACT

Generative image editing has recently witnessed extremely fast-paced growth. Some works use high-level conditioning such as text, while others use low-level conditioning. Nevertheless, most of them lack fine-grained control over the properties of the different objects present in the image, i.e. object-level image editing. In this work, we tackle the task by perceiving the images as an amalgamation of various objects and aim to control the properties of each object in a fine-grained manner. Out of these properties, we identify structure and appearance as the most intuitive to understand and useful for editing purposes. We propose **PAIR** Diffusion, a generic framework that can enable a diffusion model to control the structure and appearance properties of each object in the image. We show that having control over the properties of each object in an image leads to comprehensive editing capabilities. Our framework allows for various object-level editing operations on real images such as reference image-based appearance editing, free-form shape editing, adding objects, and variations. Thanks to our design, we do not require any inversion step. Additionally, we propose multimodal classifier-free guidance which enables editing images using both reference images and text when using our approach with foundational diffusion models. We validate the above claims by extensively evaluating our framework on both unconditional and foundational diffusion models.

## 1 INTRODUCTION

Diffusion-based generative models have shown promising results in synthesizing and manipulating images with great fidelity, among which text-to-image models and their follow-up works have great influence in both academia and industry. When editing a real image a user generally desires to have intuitive and precise control over different elements (*i.e.* the objects) composing the image in an independent manner. We can categorize existing image editing methods based on the level of control they have over individual objects in an image. One line of work involves the use of text prompts to manipulate images (Hertz et al., 2022; Liu et al., 2022; Brooks et al., 2022; Liew et al., 2022). These methods have limited capability for fine-grained control at the object level, owing to the difficulty of describing the shape and appearance of multiple objects simultaneously with text. In the meantime, prompt engineering makes the manipulation task tedious and time-consuming. Another line of work uses low-level conditioning signals such as masks Hu et al. (2022); Zeng et al. (2022b); Patashnik et al. (2023), sketches (Voynov et al., 2022), images (Song et al., 2022; Cao et al., 2023; Yang et al., 2023) to edit the images. However, most of these works either fall into the prompt engineering pitfall or fail to independently manipulate multiple objects. Different from previous works, we aim to independently control the properties of multiple objects composing an image *i.e.* object-level editing. We show that we can formulate various image editing tasks under the object-level editing framework leading to comprehensive editing capabilities.

To tackle the aforementioned task, we propose a novel framework, dubbed Structure-and-Appearance **Pair**ed **Diffusion** Models (**PAIR-Diffusion**). Specifically, we perceive an image as an amalgamation of diverse objects, each described by various factors such as shape, category, texture, illumination, and depth. Then we further identified two crucial macro properties of an object: structure and appearance. Structure oversees object's shape and category, while appearance contains details like texture, color, and illumination. To accomplish this goal, PAIR-Diffusion adopts an off-the-shelf network to estimate panoptic segmentation maps as the structure, and then extract appearance representation using pre-

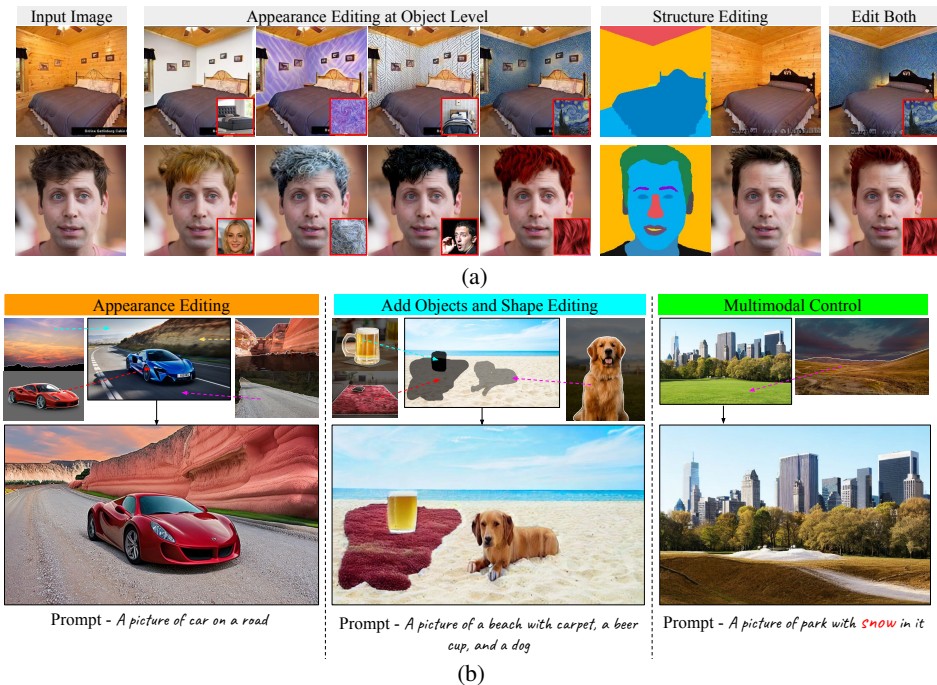

Figure 1: PAIR diffusion framework allows appearance and structure editing of an image at an object level. Our framework is general and can enable object-level editing capabilities in both (a) unconditional diffusion models and (b) foundational diffusion models. Using our framework with a foundational diffusion model allows for comprehensive in-the-wild object-level editing capabilities.

trained image encoders. We use the extracted per-object appearance and structure information to condition a diffusion model and train it to generate images. In contrast to previous text-guided image editing works (Avrahami et al., 2022; Brooks et al., 2022; Couairon et al., 2022b; Ruiz et al., 2022), we consider an additional reference image to control the appearance. Compared to text prompts although convenient, they can only vaguely describe the appearance, images can precisely define the expected texture and make fine-grained image editing easier. Having the ability to control the structure and appearance of an image at an object level gives us comprehensive editing capabilities. Using our framework we can achieve, localized free-form shape editing, appearance editing, editing shape and appearance simultaneously, adding objects in a controlled manner, and object-level image variation (Fig. 1). Thanks to our design we do not require any inversion step for editing real images.

Our approach is general and can be used with various models. In this work, we show the efficacy of our framework on unconditional diffusion models and foundational text-to-image diffusion models. We also propose multimodal classifier-free guidance to reap the full benefits of the text-to-image diffusion models. It enables PAIR Diffusion to control the final output using both reference images and text in a controlled manner hence getting best of both worlds. Thanks to our easy-to-extract representations we do not require specialized datasets for training and we show results on LSUN datasets, Celeb-HQ datasets for unconditional models and use COCO datasets for foundational diffusion models. To summarize our contributions are as follows:

- We propose PAIR-Diffusion, a general framework to enable object-level editing in diffusion models. It allows editing the structure and appearance of each object in the image independently.

- The proposed design inherently supports various editing tasks using a single model: localized free-form shape editing, appearance editing, editing shape and appearance simultaneously, adding objects in a controlled manner, and object-level image variation.

- Additionally, we propose a multimodal classifier-free guidance that enables PAIR Diffusion to edit the images using both reference images and text in a controlled manner when using the approach with foundational diffusion models.

## 2 RELATED WORKS

**Diffusion Models.** Diffusion probabilistic models (Sohl-Dickstein et al., 2015) are a class of deep generative models that synthesize data through an iterative denoising process. Diffusion models utilize a forward process that applies noise into data distribution and then reverses the forward process to reconstruct the data itself. Recently, they have gained popularity for the task of image generation (Ho et al., 2020; Song & Ermon, 2019). Dhariwal *et al.* Dhariwal & Nichol (2021) introduced various techniques such as architectural improvements and classifier guidance, that helped diffusion models beat GANs in image generation tasks for the first time. Followed by this, many works started working on scaling the models (Nichol et al., 2021; Ramesh et al., 2022a; Rombach et al., 2022b; Saharia et al., 2022) to billions of parameters, improving the inference speed (Salimans & Ho, 2022) and memory cost (Rombach et al., 2022b; Vahdat et al., 2021). LDM (Rombach et al., 2022b) is one the most popular models which reduced the compute cost by applying the diffusion process to the low-resolution latent space and scaled their model successfully for text-to-image generation trained on webscale data. Other than image generation, they have been applied to various fields such as multi-modal generation (Xu et al., 2022b),text-to-3D (Poole et al., 2022; Singer et al., 2023), language generation (Li et al., 2022), 3D reconstruction (Gu et al., 2023), novel-view synthesis (Xu et al., 2022a), music generation (Mittal et al., 2021), object detection (Chen et al., 2022), etc.

**Generative Image Editing.** Image generation models have been widely used in image editing tasks since the inception of GANs (Karras et al., 2019; Jiang et al., 2021; Gong et al., 2019; Epstein et al., 2022; Ling et al., 2021), however, they were limited to edit a restricted set of images. Recent developments in the diffusion model has enabled image editing in the wild. Earlier works (Rombach et al., 2022a; Nichol et al., 2021; Ramesh et al., 2022b) started using text prompts to control the generated image. This led to various text-based image editing works such as (Feng et al., 2022; Mokady et al., 2022; Liu et al., 2022). To make localized edits works such as (Hertz et al., 2022; Parmar et al., 2023; Tumanyan et al., 2022) use cross-attention feature maps between text and image. InstructPix2Pix (Brooks et al., 2022) further enabled instruction-based image editing. However, using only text can only provide coarse edits. Works such as (Avrahami et al., 2022; Zeng et al., 2022a) explored explicit spatial conditioning to control the structure of generated images and used text to define the appearance of local regions. Works such as (Couairon et al., 2022a; Liew et al., 2022) rely on input images and text descriptions to get the region of interest for editing. However, most of the mentioned works lack object-level editing capabilities and some still rely only on text for describing the appearance. Recent works such as (Mou et al., 2023; Epstein et al., 2023) have object-level editing capabilities, however, they are based on the classifier guidance technique at inference time which leads to limited precision. Further, they show results only on stable diffusion and require inversion to edit real images. Our framework is general and can be applied to any diffusion model. We also enable multimodal control of the appearances of objects in the image when using our framework with stable diffusion.

## 3 PAIR DIFFUSION

In this work, we aim to develop an image-editing framework that allows editing the properties of individual objects in the image. We perceive an image $x \in \mathbb{R}^{3 \times H \times W}$ as composition of objects $\mathbb{O} = \{o_1, o_2, \ldots o_n\}$ where $o_i$ represents the properties of $i^{th}$ object in the image. As discussed in Sec. 1, we focus on enabling control over the structure and the appearance of each object, hence we write $o_i = (s_i, f_i)$ where $s_i$ represents the structure, $f_i$ represents the appearance. Thus, the distribution that we aim to model can be written as

$$p(x|\mathbb{O}, y) = p(x|\{(s_1\,f_1), \ldots, (s_n,\,f_n)\}, y) \tag{1}$$

We use $y$ to represent any form of conditioning signal already present in the generative model, e.g. text, and develop our framework to enable new object-level editing capabilities while preserving the original conditioning. The rest of the method section is organized as follows. In Sec. 3.1, we describe the method to obtain $s_i$ and $f_i$ for every object in a given image. Next, in Sec. 3.2, we show that various image editing tasks can be defined in the scope of the proposed object-level formulation of images. Finally, in Sec. 3.3, we describe the usage of the representations to augment the generative models and inference techniques to achieve object-level editing in practice.

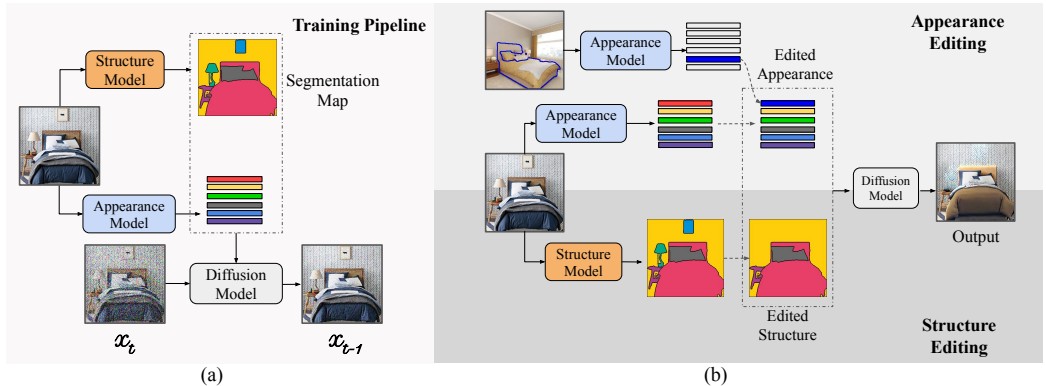

Figure 2: Overview of PAIR Diffusion. An image can be seen as a composition of objects each defined by different properties like structure (shape and category), appearance, depth, etc. We focus on controlling structure and appearance. (a) During training, we extract structure and appearance information and train a diffusion model in a conditional manner. (b) At inference, we can perform multiple editing operations by independently controlling the structure and appearance of any real image at the object level.

## 3.1 Structure and Appearance Representation

Given an image $x \in \mathbb{R}^{3 \times H \times W}$ we want to extract the structure and appearance of each object present in the image.

**Structure:** The structure oversees the object's shape and category and is represented as $s_i = (c_i, m_i)$ where $c_i$ represents the category and $m_i \in \{0, 1\}^{H \times W}$ represents the shape. We extract the structure information using a panoptic segmentation map, as it readily provides each object's category and shape information and is easy to compute. Given an off the shelf segmentation network $E_S(\cdot)$, we obtain $\boldsymbol{S} = E_S(x)$, with $\boldsymbol{S} \in \mathbb{N}^{H \times W}$ which gives direct access to $c_i, m_i$.

**Appearance:** The appearance representation is designed to capture the visual aspects of the object. To represent the object faithfully, it needs to capture both the low-level features like color, texture, etc., as well as the high-level features in the case of complex objects. To capture such a wide range of information, we choose a combination of convolution and transformer-based image encoders (Raghu et al., 2021), namely VGG (Simonyan & Zisserman, 2015) and DINOv2 (Oquab et al., 2023) to get appearance information. We use initial layers of VGG to capture low-level characteristics as they can capture details like color, texture etc. (Yosinski et al., 2015; Zeiler & Fergus, 2014). Conversely, DINOv2 has well-learned representations and has shown promising results for various downstream computer vision tasks. Hence, we use the middle layers of DINOv2 to capture the high-level characteristics of the object.

In order to compute per-object appearance representations, we first extract the feature maps from $l^{th}$ block of an encoder $E_G$, $\tilde{\mathbf{G}} = E_G^l(x)$ with $\tilde{\mathbf{G}} \in \mathbb{R}^{C \times h \times w}$, where $h \times w$ is the resulting spatial size, $C$ is the number of channels. We then parse object-level features, relying on $m_i$ to pool over the spatial dimension and obtain the appearance vector $\boldsymbol{g}_i^l \in \mathbb{R}^C$

$$\boldsymbol{g}_i^l = \frac{\sum_{j,k} E_G^l(x) \odot m_i}{\sum_{j,k} m_i} \qquad (2)$$

Here $E_G$ could be either DINOv2 or VGG. Let us use $\boldsymbol{g}_i^{Vl}$ and $\boldsymbol{g}_i^{Dl}$ to represent the appearance vectors extracted using features of VGG and DINOv2 at block $l$ respectively. The appearance information of $i^{th}$ object is given by a tuple $f_i = (\boldsymbol{g}_i^{Vl_1}, \boldsymbol{g}_i^{Dl_2}, \boldsymbol{g}_i^{Dl_3})$ where $l_2 < l_3$. The abstraction level of features in $f_i$ increases from $\boldsymbol{g}_s^{Vl_1}$ to $\boldsymbol{g}_s^{Dl_3}$.

## 3.2 IMAGE EDITING FORMULATION

We can define various image editing tasks using the proposed object-level design. Consider an image $x$ with $n$ objects $\mathbb{O} = \{o_1, o_2, \ldots o_n\}$. For each object $o_i$, we can extract $s_i, f_i$ as described in Sec. 3.1. We present fundamental image editing operations below. The editing operations can be mixed with each other enabling a wide range of editing capabilities.

**Appearance Editing** $(s_i, f_i) \to (s_i, f_i')$. It can be achieved by simply swapping appearance vector $f_i$ with an edited appearance vector $f_i'$. We can use the method described in Sec. 3.1 to extract the appearance vectors from reference images and use a convex combination of them to get $f_i'$. Formally, $f_i' = a_0 f_i + a_1 f_i^R$ where $f_i^R$ represents the appearance vectors of $i^{th}$ object in the reference image.

**Shape Editing** $(s_i, f_i) \to (s_i', f_i)$. It can be achieved by changing the structure $(c_i, m_i)$ to $(c_i, m_i')$ i.e. the shape can be explicitly changed by the user while maintaining the appearance.

**Object Addition** $\mathbb{O} \to \mathbb{O} \cup \{o_{n+1}\}$. We can add an object to an image by defining its structure and appearance. We can get them from a reference image or the user may give the structure and only appearance can come from a reference image.

**Object Appearance Variation**. We can also get object level appearance variations due to information loss when pooling features to calculate appearance vectors and the stocastic nature of the generative models.

Once we get object with edited properties $\mathbb{O}'$ and conditioning $y$ we can sample a new image from the learned distribution $p(x|\mathbb{O}', y)$. We can see that our object-level design can easily incorporate various editing abilities and help us achieve a comprehensive image editor. In the next section, we will describe a way to implement $p(x|\mathbb{O}, y)$ in practice and inference methods to sample and control the edited image.

## 3.3 ARCHITECTURE DESIGN AND INFERENCE

In practice, Eq. 1 essentially represents a conditional generative model. Given the recent success of diffusion models, we use them to realize the Eq. 1. Our extracted representations in Sec. 3.1 can be used to enable object-level editing in any diffusion model. Here we briefly describe a method to use our represents on the unconditional diffusion models and foundational text-to-image (T2I) diffusion model. We start by representing structure and appearance in a spatial format to conveniently use them for conditioning. We represent the structure conditioning as $\mathbf{S} \in \mathbb{N}^{2 \times H \times W}$ where the first channel has category information and the second channel has the shape information of each object. For appearance conditioning, we first $L2$ normalize each vector along channel dimension and splat them spatially using $m_i$ and combine them in a single tensor represented as $\mathbf{G} \in \mathbb{R}^{C \times H \times W}$ which leads to $F = (\mathbf{G}^{Vl_1}, \mathbf{G}^{Dl_2}, \mathbf{G}^{Dl_3})$. We then concatenate $\mathbf{S}$ to every element of $F$ which results in our final conditioning signals $F_s = (\mathbf{G}_s^{Vl_1}, \mathbf{G}_s^{Dl_2}, \mathbf{G}_s^{Dl_3})$.

In the case of the foundational T2I diffusion model, we choose Stable Diffusion (SD) (Rombach et al., 2022a) as our base model. In order to condition it, we adopt ControlNet (Zhang & Agrawala, 2023) because of its training and data efficiency in conditioning SD model. The control module consists of encoder blocks and middle blocks that are replicated from SD UNet architecture. There are various works that show in SD the inner layers with lower resolution tend to focus more on high-level features, whereas the outer layers focus more on low-level features (Cao et al., 2023; Tumanyan et al., 2022; Liew et al., 2022). Hence, we use $\mathbf{G}_s^{Vl_1}$ as input to the control module and add $\mathbf{G}_s^{Dl_2}, \mathbf{G}_s^{Dl_3}$ to the features after cross-attention in first and second encoder blocks of the control module respectively. For the unconditional diffusion model, we use the unconditional latent diffusion model (LDM) (Rombach et al., 2022a) as our base model. Pertaining to the simplicity of the architecture and training of these models we simply concatenate the features in $F_s$ to the input of LDM. The architecture is accordingly modified to incorporate the increased number of input channels. For further details please refer to *Supp. Mat.* .

For training both the models we follow standard practice (Rombach et al., 2022a) and use the simplified training objective $\mathcal{L} = ||\epsilon - \epsilon_\theta(z_t, \mathbf{S}, F, y, t)||_2^2$ where $z_t$ represents the noisy version of $x$ in latent space at timestep $t$ and $\epsilon$ is the noise used to get $z_t$ and $\epsilon_\theta$ represents the model being trained.

In the case of Stable Diffusion, $y$ represents the text prompts, whereas $y$ can be ignored in the case of the unconditional diffusion model.

**Multimodal Inference.** Having a trained model we need an inference method such that we can guide the strengths of various conditioning signals and control the edited image. Here, we take the case when $y$ is text and unconditional diffusion models would be a special case where $y$ is null. Specifically, the structure **S** and appearance $F$ come from a reference image and the information in $y$ could be disjoint from $F$, we need a way to capture both in the final image. A well-trained diffusion model estimates the score function of the underlying data distribution Song et al. (2020) *i.e* $\nabla_{z_t} p(z_t | \mathbb{O}, y) = \nabla_{z_t} p(z_t | \mathbf{S}, F, y)$, which in our case can be expanded as

$$\nabla_{z_t} \log p(z_t | \mathbf{S}, F, y) = \nabla_{z_t} \log p(z_t | \mathbf{S}, F) + \nabla_{z_t} \log p(z_t | y) - \nabla_{z_t} \log p(z_t) \tag{3}$$

We use the concept of classifier-free guidance (CFG) Ho & Salimans (2022) to represent all score functions in the above equation using a single model by dropping the conditioning with some probability during training. Using the CFG formulation we get the following update rule from Eq. 3

$$\tilde{\epsilon}_\theta(z_t, \mathbf{S}, F, y) = \epsilon_\theta(z_t, \phi, \phi, \phi) + s_S \cdot (\epsilon_\theta (z_t, \mathbf{S}, \phi, \phi) - \epsilon_\theta (z_t, \phi, \phi, \phi))$$
$$+ s_F \cdot (\epsilon_\theta (z_t, \mathbf{S}, F, \phi) - \epsilon_\theta (z_t, \mathbf{S}, \phi, \phi)) + s_y \cdot (\epsilon_\theta (z_t, \phi, \phi, y) - \epsilon_\theta (z_t, \phi, \phi, \phi)) \tag{4}$$

For brevity, we did not include $t$ in the equation above. A formal proof of the above equations is provided in *Supp. Mat.* . Intuitively, $F$ is more information-rich compared to $y$. Due to this, during training the network learns to give negligible importance to $y$ in the presence of $F$ and we need to use $y$ independently of $F$ during inference to see its effect on the final image. In Eq. 4 $s_S, s_F, s_y$ are guidance strengths for each conditioning signal. It provides PAIR Diffusion with an intuitive way to control and edit images using various conditions. For example, if a user wants to give more importance to a text prompt compared to the appearance from the reference image, it can set $s_y > s_F$ and vice-versa. For the unconditional diffusion models, we can simply ignore the term corresponding to $s_y$ in Eq 4.

## 4 EXPERIMENTS

In this section, we present qualitative and quantitative analysis that show the advantages of the PAIR diffusion framework introduced in Sec. 3. We refer to UC-PAIR Diffusion to denote our framework applied to unconditional diffusion models and reserve the name PAIR Diffusion when applying the framework to Stable Diffusion. Evaluating image editing models is hard, moreover, there are few works that have comprehensive editing capabilities at the object level making a fair comparison even more challenging. For these reasons, we perform two main sets of experiments. Firstly, we train UC-PAIR Diffusion on widely used image-generation datasets such as the bedroom and church partitions of the LSUN Dataset (Yu et al., 2015), and the CelebA-HQ Dataset (Karras et al., 2017). We conduct quantitative experiments on these datasets as they represent a well-study benchmark, with a clear distinction between training and testing sets, making it easier and fairer to perform evaluations. Secondly, we fine-tune PAIR Diffusion on the COCO (Lin et al., 2014) dataset. We use this model to perform in-the-wild editing and provide examples for the use cases described in Sec. 3.2, showing the comprehensive editing capabilities of our method. We refer the reader to the *Supp. Mat.* for the details regarding model training and implementations, along with additional results.

### 4.1 EDITING APPLICATIONS

In this section, we qualitatively validate that our model can achieve comprehensive object-level editing capabilities in practice. We primarily show results using PAIR Diffusion and refer to the *Supp. Mat.* for results on smaller datasets. We use different baselines according to the editing task. We adapt Prompt-Free-Diffusion (PFD) Xu et al. (2023) as a baseline for localized appearance editing, by introducing masking and using the cropped reference image as input. Moreover, we adopt Paint-By-Example (PBE) Yang et al. (2023) as a baseline for adding objects and shape editing. For further details regarding implementation please refer to *Supp. Mat.* . When we want the final output to be influenced by the text prompt as well we set $s_y > s_F$ else we set $s_y < s_F$. For the figures

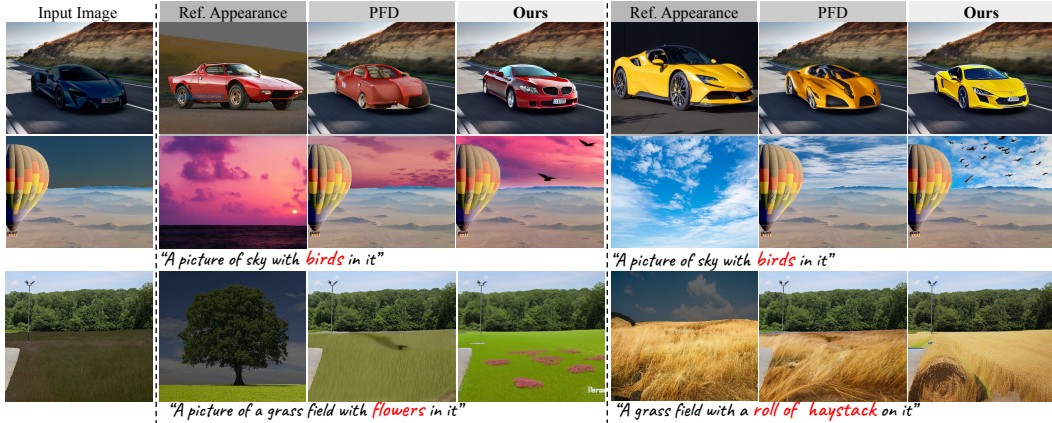

Figure 3: Qualitative results for appearance editing.

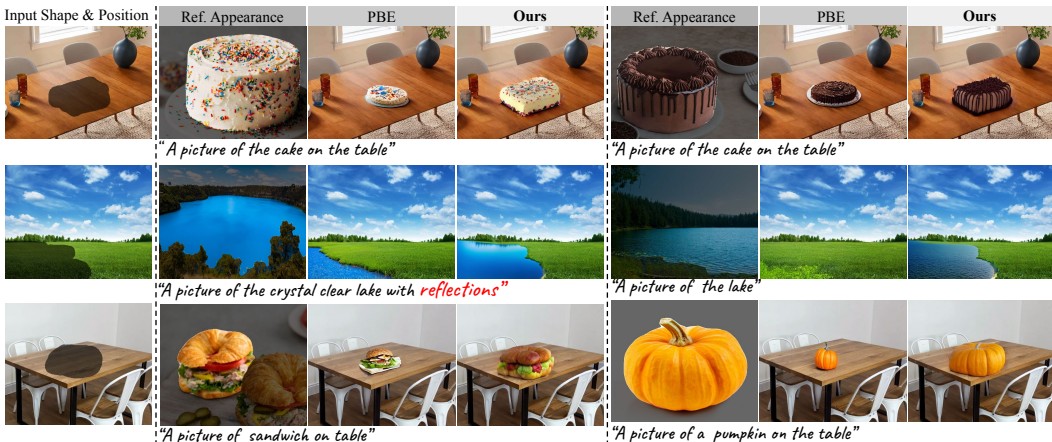

Figure 4: Qualitative results for adding objects and shape editing.

where there is no prompt provided below the image assume that prompt was auto-generated using the template *A picture of {category of object being edited}*. When editing a local region we used a masked sampling technique to only affect the selected region (Rombach et al., 2022a).

**Appearance Editing.** In Fig. 3, we report qualitative results for appearance editing driven by reference images and text. We can see that our multilevel appearance representation and object-level design help us edit the appearance of both simple objects such as the sky as well as complex objects like cars. On the other hand, PFD (Xu et al., 2023) gives poor results when editing the appearance of complex objects due to the missing object-level design. Furthermore, using our multimodal classifier free guidance, our model can seamlessly blend the information from the text and the reference images to get the final edited output whereas PFD (Xu et al., 2023) lacks this ability.

**Add objects and Shape editing.** We show the object addition and shape editing operations result together in Fig. 4. With PAIR Diffusion we can add complex objects with many details like a cake, as well as simpler objects like a lake. When changing the structure of the cake from a circle to a square, the model captures the sprinkles and dripping chocolate on the cake while rendering it in the new shape. In all the examples, we can see that the edges of the newly added object blend smoothly with the underlying image. On the other hand, PBE (Yang et al., 2023) completely fails to follow the desired shape and faces issues with large objects like lakes.

**Variation.** We can also achieve image variations at an object level as shown in Fig. 13 in *Supp. Mat.* . We can see that our model can capture various details of the original object and still produce variations.

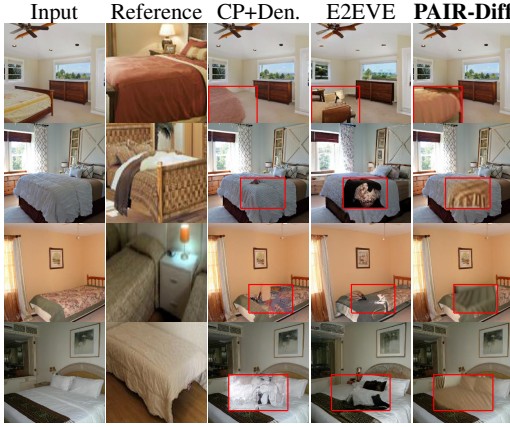

Input    Reference    CP+Den.    E2EVE    **PAIR-Diff**

Table 1: Quantitative results for appearance control on the LSUN Bedroom validation set.

| Model | FID ($\downarrow$) | L1 ($\downarrow$) | SSIM ($\uparrow$) |
|---|---|---|---|
| Copy-Paste (CP) | 21.37 | 0.0 | 0.87 |
| Inpainting | 8.25 | 0.02 | 0.17 |
| CP+Denoise | 9.15 | 0.02 | 0.32 |
| E2EVE | 13.59 | 0.05 | 0.34 |
| **PAIR-Diff** | 12.81 | 0.02 | 0.51 |

Table 2: Quantitative results for structure control on CelebA-HQ validation dataset.

| Model | mIoU ($\uparrow$) | SSIM ($\uparrow$) |
|---|---|---|
| SEAN | 0.64 | 0.32 |
| **PAIR-Diff** | 0.67 | 0.52 |

Figure 5: Visual results for appearance control on LSUN bedroom. We show the results obtained with relevant baselines for editing the red area in the input image using the reference as a driver.

## 4.2 QUANTITATIVE RESULTS

As described in Sec.3, the backbone of our design is the ability to control two major properties of the objects, the appearance and the structure. The aim of the quantitative evaluation is to verify that we can control the mentioned properties and not necessarily to push the state-of-the-art results. We start by evaluating our model on appearance control: the task consists of modifying a specific region of the input image using a reference image to drive the edit. We compare our method with the recent work of Brown et al. (2022) (E2EVE), and follow their evaluation procedure. In particular, different models are compared based on: (i) Naturalness: we expect the edited image to look realistic and rely on FID between input and edited images to assess it, (ii) Locality: we expect the edit to be limited to the specific region where the edit is performed and use L1 distance to measure it, (iii) Faithfulness: we expect the edited region and the target image to be similar and we use SSIM to evaluate it. As discussed in E2EVE, all the above-mentioned criteria should hold at the same time, and the best-performing method is the one giving good results in the three metrics at the same time. We compare our method with four baselines: (1) Copy-Paste: the driver image is simply copied in the edit region of the input image, (2) Inpainting: we use LDM Rombach et al. (2022b) to inpaint the target edit region, (3) Copy-Paste + Denoise: starting form copy-paste edit, we invert the image with DDIM, and denoise it with LDM, (4) E2EVE. In Tab. 1 we report the quantitative results on the validation set of LSUN Bedroom (Yu et al., 2015) and visual comparisons are shown in Fig.5. The copy-paste baseline provides an upper bound to the faithfulness and locality but produces images that are unrealistic (high FID score). Vice-versa, Inpainting and CP+Denoise produce natural results (low FID score) but are not faithful to the driver image (low SSIM score). Only our method performs well w.r.t. all the aspects and outperforms E2EVE in all metrics showing that we can control the appearance of a region. We refer the reader to *Supp. Mat.* for a detailed description of the evaluation procedure and baseline implementation.

Secondly, we evaluate the structure-controlling ability of our method. We adopt the validation set of CelebA-HQ (5000 samples) and compare with SEAN (Zhu et al., 2020). We generate images conditioning the model on the ground truth structure maps from the validation set and then segment the generated images with a pre-trained model zllrunning (2019). We report the mIoU score, calculated using the ground truth segmentation map as the reference, as well as the SSIM score in Table 2. The proposed method outperforms Zhu et al. (2020) in terms of both mIoU and SSIM, demonstrating that our method can precisely follow the guidance of structure and retain the appearance.

## 4.3 ABLATION STUDY

**Multimodal Classifier Free Guidance.** Here we validate the effectiveness of the proposed multimodal classifier-free guidance. Instead of factorizing which results in Eq. 3 we directly expand the conditional score function $\nabla_{z_t} \log p(z_t|\mathbf{S}, F, y)$ and apply classifier free guidance formulation on it and get the following equation.

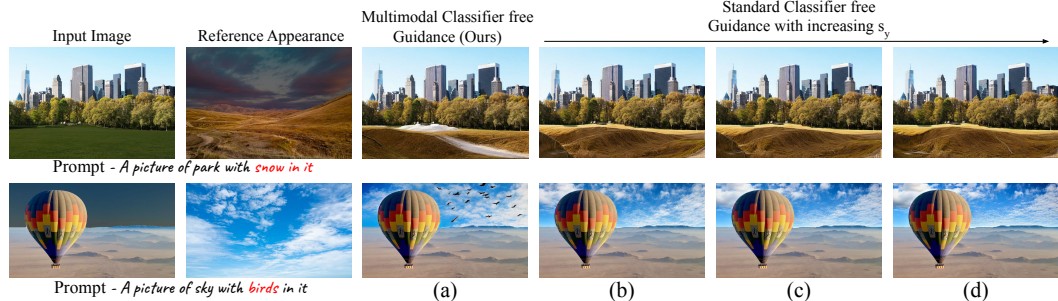

Figure 6: Ablation study for multimodal classifier-free guidance. We can see that if we use standard classifier-free guidance (Eq. 5) the model completely ignores the text when sampling the image.

$$\tilde{\epsilon}_\theta(z_t, \mathbf{S}, F, y) = \epsilon_\theta(z_t, \phi, \phi, \phi) + s_S \cdot (\epsilon_\theta(z_t, \mathbf{S}, \phi, \phi) - \epsilon_\theta(z_t, \phi, \phi, \phi))$$
$$+ s_F \cdot (\epsilon_\theta(z_t, \mathbf{S}, F, \phi) - \epsilon_\theta(z_t, \mathbf{S}, \phi, \phi)) + s_y \cdot (\epsilon_\theta(z_t, \mathbf{S}, F, y) - \epsilon_\theta(z_t, \mathbf{S}, F, \phi)) \quad (5)$$

We highlight the difference between Eq. 4 and Eq. 5 using blue color. We compare the results sampled from Eq. 4 and Eq. 5 in Fig. 6. In the figure, column (a) shows results from Eq. 4 whereas column (b) - (d) shows results from Eq. 5 with increasing $s_y$. We use the same seed to generate all the images, further the values of $s_S, s_F, s_y$ are the same in columns (a) and (b). For the first row we set $s_S = 8, s_F = 3, s_y = 8$ and for second row it is $s_S = 6, s_F = 4, s_y = 8$. The values of $s_y$ for (b) - (d) are 8, 15, 20 respectively. We can clearly see that sampling results using Eq. 5 completely fail to take text prompt into consideration even after increasing the value of $s_y$. This shows the effectiveness of the proposed classifier-free guidance Eq. 4 for controlling the image in a multimodal manner.

**Appearance representation.** We ablate the importance of using both VGG and DINOv2 for representing the appearance of an object. We train two models, one using only VGG features for appearance and the second using only DINOv2 features for appearance and are represented using $M_{VGG}$ and $M_{DINO}$ respectively. We trained two models using exactly the same hyperparameters as our original model. We validate it using pairwise image similarity metrics on the COCO validation dataset. We use L1 as our low-level metric and LPIPS (Zhang et al., 2018) as our high-level metric, results are shown in Tab. 3. We can see that $M_{VGG}$ has a better L1 score compared to $M_{DINO}$ whereas LPIPs score is better for $M_{DINO}$ compared to $M_{VGG}$. This shows that VGG features are good at capturing low-level details and DINO features are good at capturing high-level details in our design. When we use both features together we get the best L1 and LPIPS score getting the best of both of the features. Hence, in our final design, we used both VGG and DINOv2 features for appearance vectors. Supporting visuals can be found in Fig 9.

Table 3: Quantitative results of ablation study on appearance representation

| Model | L1 $\downarrow$ | LPIPS $\downarrow$ |
|---|---|---|
| $M_{VGG}$ | 0.1893 | 0.555 |
| $M_{DINO}$ | 0.1953 | 0.549 |
| Ours | **0.1891** | **0.545** |

## 5  CONCLUSION

In this paper, we showed that we can build a comprehensive image editor by leveraging the observation that images are amalgamations of various objects. We proposed a generic framework dubbed PAIR Diffusion that enables structure and appearance editing at object-level editing in any diffusion model. Our proposed framework enables various object-level editing operations on real images without the need for inversion such as appearance editing, structure editing, adding objects, and variations all of which can be done by training the model only once. We also proposed multimodal classifier-free guidance which enables multimodal control in the editing operations when using our framework with models like stable diffusion. We validated the efficacy of the framework by showing extensive editing results using our model on diverse domains of real images. In the future, one can also explore the design of appearance vectors such that we can further control the illumination, pose, etc, and have better identity preservation of the object being edited. We hope that our work motivates future works to move in the direction of object-level image editing which might help to formulate and build an all-in-one image editing model.

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
