APPENDIX

This appendix contains three sections organized as follows. In Sec.A we discuss our proposed multimodal classifier free guidance, proving Eq. 4 of the main paper and providing qualitative results for its usage. In Sec. B, we report the implementation details of our unconditional model along with details about the baselines adopted in the paper. Lastly, in Sec. C we show additional qualitative results and applications of our framework.

## A  MULTIMODAL CLASSIFIER FREE GUIDANCE

### A.1  PROOF

Let us represent $p(z|c)$ as the distribution we want to learn where $c$ is $\mathbf{S}, F, y$. Here $z$ represents $x$ in latent space. A well-trained diffusion model learns to estimate $\nabla_{z_t} \log p(z_t|c)$. Let us focus on $\log p(z_t|c)$. Using Bayes's rule and taking log

$$\log p(z_t|c) = \log p(c|z_t) + \log p(z_t) - \log p(c) \tag{6}$$

In our setting of multimodal inference, we want to affect the final image using the appearance from the reference image $F$ and the text prompt $y$ independently. Multimodal inference would be particularly useful when the information in $y$ would be disjoint from $\mathbf{S}, F$ and we need to capture it in the final image. Using this we can write.

$$\log p(\mathbf{S}, F, y) = \log p(\mathbf{S}, F) + \log p(y) \tag{7}$$

Now expanding $\log p(c|z_t)$ and using Eq 7

$$\begin{aligned} \log p(c|z_t) &= \log p(\mathbf{S}, F, y|z_t) \\ \log p(\mathbf{S}, F, y|z_t) &= \log p(\mathbf{S}, F|z_t, y) + \log p(y|z_t) \\ \log p(\mathbf{S}, F, y|z_t) &= \log p(\mathbf{S}, F|z_t) + \log p(y|z_t) \end{aligned} \tag{8}$$

Plugging the Eq. 8 in Eq. 6 and using Eq. 7 again we get

$$\begin{aligned} \log p(z_t|\mathbf{S}, F, y) &= \log p(\mathbf{S}, F|z_t) + \log p(y|z_t) + \log p(z_t) - \log p(\mathbf{S}, F, y) \\ \log p(z_t|\mathbf{S}, F, y) &= \log p(\mathbf{S}, F|z_t) + \log p(y|z_t) + \log p(z_t) - \log p(\mathbf{S}, F) - \log p(y) \\ \log p(z_t|\mathbf{S}, F, y) &= \log p(z_t|\mathbf{S}, F) + \log p(z_t|y) - \log p(z_t) \end{aligned} \tag{9}$$

In Eq. 8 and Eq. 9 we used the observation that Taking gradient w.r.t. $\nabla_{z_t}$ we can get

$$\nabla_{z_t} \log p(z_t|\mathbf{S}, F, y) = \nabla_{z_t} \log p(z_t|\mathbf{S}, F) + \nabla_{z_t} \log p(z_t|y) - \nabla_{z_t} \log p(z_t) \tag{10}$$

Applying baye's rule to $\nabla_{z_t} \log p(z_t|\mathbf{S}, F)$ and $\nabla_{z_t} \log p(z_t|y)$

$$\nabla_{z_t} \log p(z_t|\mathbf{S}, F) = \nabla_{z_t} \log p(F|\mathbf{S}, z_t) + \nabla_{z_t} \log p(S|z_t) + \nabla_{z_t} \log p(z_t) - \underbrace{\nabla_{z_t} \log p(\mathbf{S}, F)}_{0} \tag{11}$$

$$\nabla_{z_t} \log p(z_t|y) = \nabla_{z_t} \log p(y|z_t) + \nabla_{z_t} \log p(z_t) - \underbrace{\nabla_{z_t} \log p(y)}_{0} \tag{12}$$

Pluggin Eq. 11 and Eq. 12 in Eq. 10 we get

$$\nabla_{z_t} \log p(z_t|\mathbf{S}, F, y) = \nabla_{z_t} \log p(F|\mathbf{S}, z_t) + \nabla_{z_t} \log p(S|z_t) + \nabla_{z_t} \log p(y|z_t) + \nabla_{z_t} \log p(z_t) \tag{13}$$

We can use the concept of classifier-free-guidance Ho & Salimans (2022) to approximate the above equation using a single model which has been trained by dropping the conditions during training and get the final sampling equation Eq. 4

$$\tilde{\epsilon}_\theta(z_t, \mathbf{S}, F, y) = \epsilon_\theta(z_t, \phi, \phi, \phi) + s_S \cdot (\epsilon_\theta(z_t, \mathbf{S}, \phi, \phi) - \epsilon_\theta(z_t, \phi, \phi, \phi))$$
$$+s_F \cdot (\epsilon_\theta(z_t, \mathbf{S}, F, \phi) - \epsilon_\theta(z_t, \mathbf{S}, \phi, \phi)) + s_y \cdot (\epsilon_\theta(z_t, \phi, \phi, y) - \epsilon_\theta(z_t, \phi, \phi, \phi)) \tag{14}$$

### A.2 ABLATION ON CFG CONTROL PARAMETERS

The multimodal classifier free guidance has three control parameters namely $s_S, s_F, s_y$. In practice, the values $s_S, s_F, s_y$ can be understood as the guidance strengths to control how the final image is affected by the structure, reference image, and the given text prompt. When using it to edit real-world images it is crucial to understand how they change the output image when varied together. In this section, we study the effect of (a) varying structure guidance $s_S$ and reference image guidance $s_F$ (b) varying text prompt guidance $s_y$ and reference image guidance $s_F$.

**Structure ($s_s$) and Appearance ($s_F$).** We explain the importance of parameters by choosing a reference image that is completely different from the underlying structure in the input image. The results are shown in Fig. 7. We can observe that as we increase $s_F$ the model forcefully imposes the reference image appearance on the object being edited and the object starts losing its structural integrity. We can get back the structural integrity when increasing $s_S$ as well. Notice, in the second row, when $s_F = 4$ and $s_S = 2$ we cannot see any parts of the car such as the wheel, headlight, windshield, etc. When we start increasing $s_S$ the subpart of the car starts appearing the edited region starts looking more like a car. However, when we increase $s_F$ too much as in the last row, even after increasing $s_S$ does not help much. In general, it is good practice to keep $s_S > s_F$ when editing real images. Further, we can adjust $s_F$ how closely we want the edited output to follow the reference appearance.

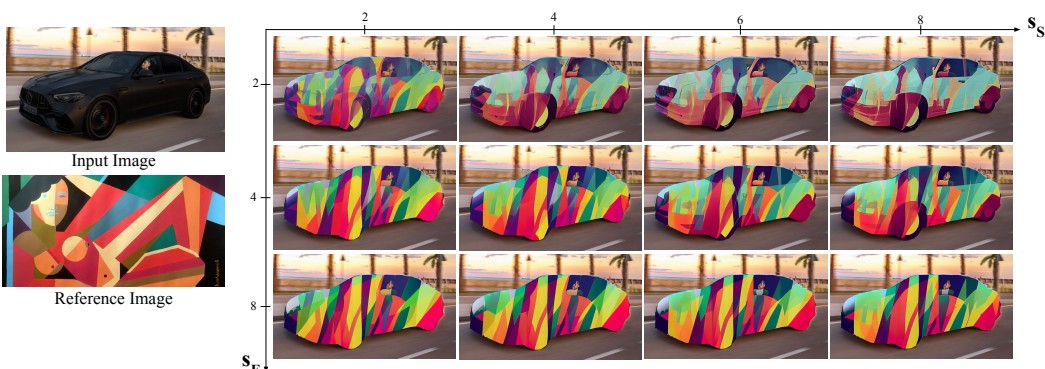

Figure 7: The figure shows the affect of $s_F$ and $s_S$ parameter in multimodal classifier free guidance Eq. 4 when they are varied. We can see that as we increase $s_F$ the output forcefully imposes the appearance from the reference image and the car starts losing the structure details that were helping to make it look like a car. We need to increase $s_S$ as well to maintain the structural integrity of the car. However, when we increase $s_F$ too much even after increasing $s_S$ does not help much.

**Text Prompt ($s_y$) and Appearance ($s_F$):** Here we try to analyze the effect of two crucial guidance parameters which help us in controlling and editing images in a multimodal manner. The results are shown in Fig 8. We can see that it is crucial to keep $s_y > s_F$ to see the effects of the text prompt on the output image. Further, we can see that the model is more sensitive to the $s_F$ parameter compared to $s_y$. Even if we increase $s_F$ slightly we can see diminishing effects of the prompt on the edited image. In general when editing images in a multimodal manner it is a good practice to keep $s_y > s_F$

and $s_F$ should have a low value in an absolute sense. Keeping these constraints we can vary the parameters to adjust the final output as per the need.

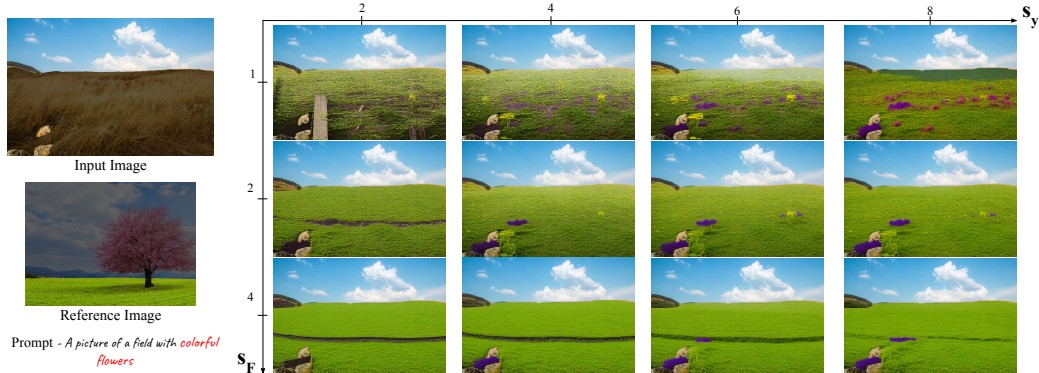

Figure 8: The figure shows the effect of $s_F$ and $s_T$ parameter in multimodal classifier free guidance Eq. 4 when they are varied. $s_F$ controls the effect of the reference image and $s_y$ control the effect of the prompt. We can see that the output is highly sensitive to $s_F$. Even if we increase $s_F$ slightly the effect of the prompt starts disappearing. When we want to use both reference images and prompts to affect the final image it is best to keep $s_y > s_F$ and $s_F$ should have a low value in an absolute sense.

# B  IMPLEMENTATION DETAILS

## B.1  PAIR DIFFUSION

**Unconditional Diffusion Model.** In this section, we provide additional details for implementing PAIR Diffusion framework on unconditional diffusion models. We used LDM Rombach et al. (2022a) as our base architecture and trained on LSUN Church, Bedroom, and CelebA-HQ datasets. To extract the structure information, we apply SeMask-L Jain et al. (2021) with Mask2former Cheng et al. (2022) trained on ADE20K Zhou et al. (2017), and compute the segmentation mask for LSUN Church and Bedroom datasets Yu et al. (2015). In CelebA-HQ, ground truth segmentation masks are available. Pertaining to the simplicity of the architecture and training datasets of these models we found that simply using $\mathbf{G}^{Vl_1}$ as appearance vectors is sufficient to achieve various editing capabilities. For conditioning, we simply concatenate $\mathbf{G}^{Vl_1}$ to the noisy latent $z_t$ along the channels dimension. We increase the number of channels of the first convolutional layer of the U-Net from $C_{in}$ to $C_{in} + C + 1$ and keep the rest of the architecture as in Rombach et al. (2022b). Here $C$ is the number of channels in $\mathbf{G}^{Vl_1}$ where $l_1 = 1$. For all three datasets namely, LSUN Church, Bedroom, and CelebA-HQ we start with the pre-trained weights provided by LDM Rombach et al. (2022b) and finetune with exactly the same hyperparameters mentioned in the paper Rombach et al. (2022b). The number of steps, learning rate, and batch size are reported in Tab 4. We train our models using A100 GPUs. During training, we randomly dropped structure and appearance conditioning with a probability of $10\%$. At inference time, we adapt Eq. 4 for sampling from the model by setting $s_y = 0$ and use classifier-free guidance style sampling using the DDIM algorithm Song & Ermon (2019) with 250 steps.

**Foundational Diffusion Model.** We use Stable Diffusion (SD) Rombach et al. (2022a) as our base architecture and ControlNet Zhang & Agrawala (2023) to efficiently condition SD. We use

| Base Model | Dataset | lr | batch size | Iterations | GPU Days |
|---|---|---|---|---|---|
| LDM | Bedroom | 9.6e-5 | 48 | 750k | 24 |
| LDM | Churches | 1.0e-5 | 96 | 350k | 12 |
| LDM | CelebA-HQ | 9.6e-5 | 24 | 120k | 1 |
| SD + ControlNet | COCO | 1.5e-5 | 128 | 86k | 96 |

Table 4: Hyperparameters for PAIR diffusion when used with LDM Rombach et al. (2022a) and SD + ControlNet Zhang & Agrawala (2023).

Figure 9: Object level appearance variations are shown for models trained with different appearance representations. It can be observed that $M_{DINO}$ faces difficulty in preserving the color when generating variations. The sky is a bit darker than the input image whereas, in the case of the couch, the shade of orange color does not match the input image. We can see that $M_{VGG}$ is able to preserve the color however it can have artifacts even for slightly complex objects. In the case of the couch, we can see some white artifacts that might have come from the white bottom of the couch in the input image. $M_{VGG}$ have a poor understanding of objects compared to $M_{DINO}$. Only the full model which uses both captures the visual aspects of the object faithfully.

COCO Caesar et al. (2018) dataset for the experiments and it already contains panoptic segmentation masks and image captions. In vanilla ControlNet, the conditioning signal is passed through a zero convolution network and added to the input noise after being passed through the first encoder block of the control module. We use $\mathbf{G}_s^{Vl_1}$ in a similar manner and use it as input to the control module as in vanilla ControlNet. Further, we modulate the features in the first encoder block of the control module using $\mathbf{G}^{Dl_2}$ and in the second encoder block using $\mathbf{G}^{Dl_3}$ by adding them to the respective features after cross-attention blocks. Before adding $\mathbf{G}^{Dl_2}, \mathbf{G}^{Dl_3}$ we pass them through two linear layers to match the dimension of the features of the network. We train the network as described in ControlNet Zhang & Agrawala (2023) and only train the control module and the linear layers, while all the other parameters are frozen. We found that the combination $l_1 = 1, l_2 = 6, l_3 = 18$ worked best after performing grid search. During training, we randomly dropped structure, appearance, and text conditioning with a probability of $10\%$. Our model is trained across 4 A100 machines with 8 GPUs each and took 3 days to train the model. The number of steps, learning rate, and batch size are reported in Tab 4. At inference time, we apply Eq. 4 for sampling and use classifier-free guidance style sampling using the DDIM algorithm Song & Ermon (2019) with 20 steps.

## B.2    BASELINES

**Quantitative Experiments.** We provide additional details about implementation and evaluation procedures for the results shown in Sec. 4.2 of the main paper. We evaluate the models on the task of in-domain appearance manipulation. We first describe the data-collection procedure. We use 5000 images from the validation set of LSUN Bedroom, and choose the bed as the object to edit. For each image, we randomly select a patch within the bed, and use a patch extracted in the same way from another image as the driver for the edit. Next, we describe the baselines. **Copy-Paste**: The target patch is copied and pasted in the target region of the input image, resizing the patch to fit the target region. **Inpainting**, we use the model pretrained on LSUN Bedroom Dataset by Rombach et al. (2022b), and use it to inpaint the edit region. To do that, we use a masked sampling technique, as done in the inpainting task in  Rombach et al. (2022a). **CP+Denoise**, we start from the results of copy-paste and apply DDIM Inversion to map the image to the diffusion noise space Song & Ermon (2019). Subsequently, we apply LDM to denoise the image to the final result. Lastly, we compare our method with **E2EVE** Brown et al. (2022). We use the original pre-trained weights shared by the authors and use their model to perform the edit. Next, we detail the metrics calculation pipeline. We compute *naturalness* by measuring the FID between the edited images and the original images from the whole dataset. We estimate the *locality*, by measuring the L1 loss between the original image and the edited image outside the edited region (i.e. the region that should not change). Finally, the *faithfulness* is measured by the SSIM between the driver image and the edited region in the edited image.

**Qualitative Experiments.** We run Prompt-Free-Diffusion Xu et al. (2023) following the description in Sec. 4.5 of the paper and the github issue https://github.com/SHI-Labs/Prompt-Free-Diffusion/issues/3#issuecomment-1573091747. Specifically, we crop the input image around the object being edited and the reference image around the selected object. We feed the two crops to the SeeCoder, and use the segmentation ControlNet with the segmentation map of the cropped input

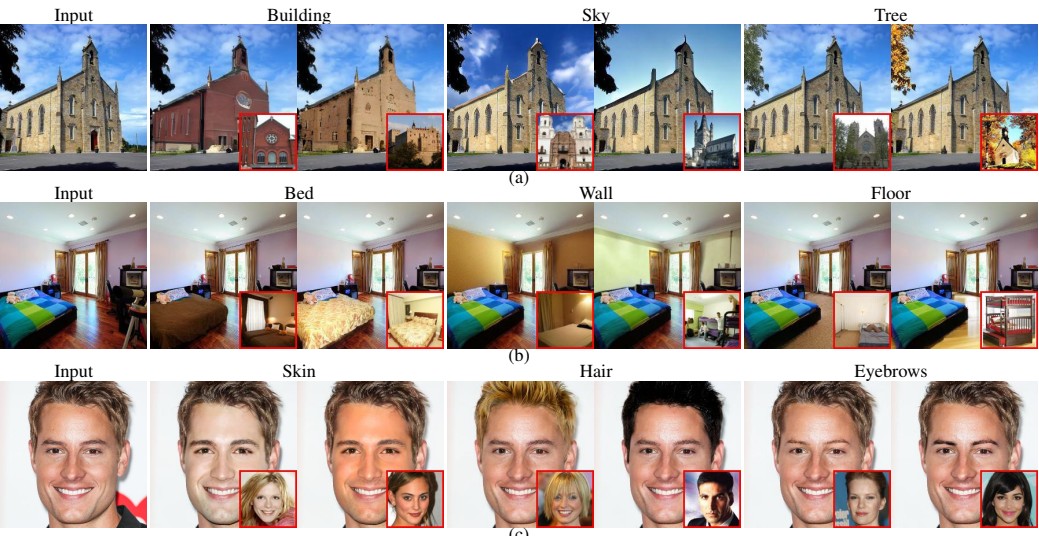

Figure 10: Qualitative results on In Domain Appearance Manipulation. Reference images are shown in the bottom right, while the text on top indicates the object targeted by the editing operation.

image as the conditioning signal. We then get the edited image by cutting the region of interest in the output image and pasting it in the input image. Regarding Paint-By-Example Yang et al. (2023), we crop the reference image around the selected object and follow the original inference procedure afterward.

## C QUALITATIVE RESULTS

### C.1 STABLE DIFFUSION RESULTS

In this section, we show additional results for PAIR-Diffusion when coupled with a foundational diffusion model like Stable Diffusion Rombach et al. (2022b). In Fig. 11, we perform appearance manipulation in the wild, showing realistic edits in different scenarios. In Fig. 12, we provide additional results for the task of adding a new object to a given scene. Lastly, we showcase another capability of our model, producing variations of a given object. Specifically, we sample different initial latent codes $z_i \sim \mathcal{N}(0, 1)$, while fixing the structure and appearance representation. We report the results in Fig. 13.

### C.2 UNCONDITIONAL PAIR DIFFUSION

We start by providing additional results for the task of appearance control. In Fig. 10 (a), we can notice that our method can easily transfer the appearance of a church from a completely different structure in the reference image to the structure of the church in the input image. At the same time, we can copy relatively homogeneous regions like the sky, transferring the color accurately, as well as more textured objects such as the trees. In Fig. 10 (b), it is interesting to note that, when we change the style of the floor, the model can appropriately place the reflections hence realistically harmonizing the edit with the rest of the scene. Similar observations can be made when we edit the wall. Lastly, in Fig. 10 (c) we show results on faces. We can observe that our method accurately transfers the appearance from the reference image, modifying the skin, hair, and eyebrows of the input. We notice that all our edits do not alter the identity of the person in the input image, which is a desirable property when editing faces.

Next, we provide an additional qualitative comparison for the task of appearance control with the baselines detailed in Sec.B. In Fig. 14, we can observe that our method seamlessly transfers the appearance from the reference image to the input image, while maintaining the edit to the targeted region. Moreover, we show the qualitative comparison with SEAN Zhu et al. (2020) in Fig. 15. We

can see that our method gives better editing results and we also allow to control the strength of the edit.

In Fig. 16, we showcase more nuanced appearance editing results instead of simply swapping the appearance of input and reference images (i.e. $f'_i = f^R_i$) by linearly combining the two. We exploit the flexibility of our formulation by setting $a_0 = \lambda, a_1 = 1 - \lambda$, with $\lambda \in [0, 1]$, *i.e.* interpolating input and reference images. We can notice how the appearance of the edited region smoothly transitions from the original appearance to the reference, providing an additional level of control for the end user.

Lastly, we present a more challenging editing scenario using reference images that contain no semantics (e.g. abstract paintings) and use it to perform both localized and global editing in Fig. 17.

| Input | Reference Appearance | **PAIR-Diffusion** |
|---|---|---|

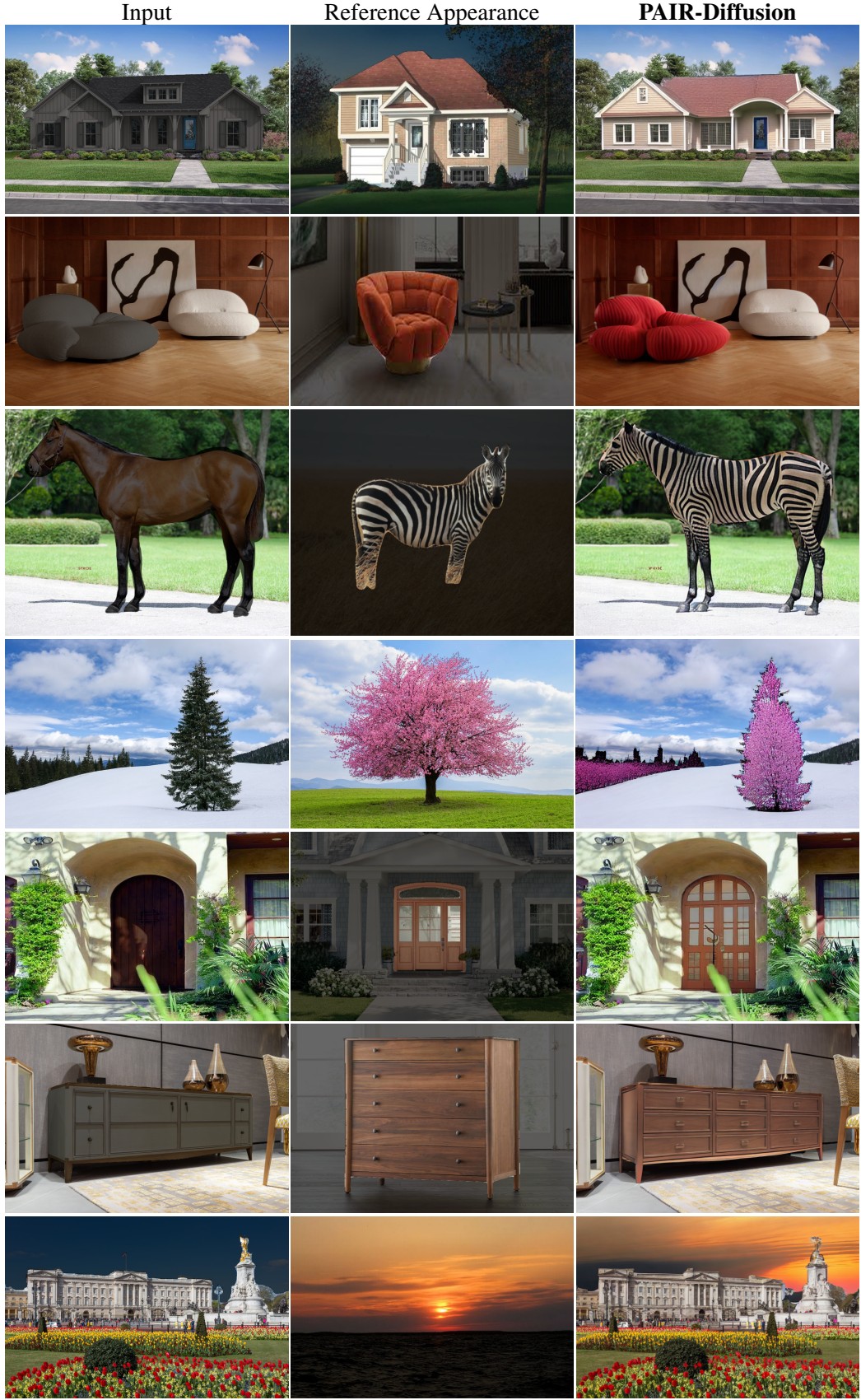

Figure 11: Appearance editing in the wild.

Input          Reference Appearance          **PAIR-Diffusion**

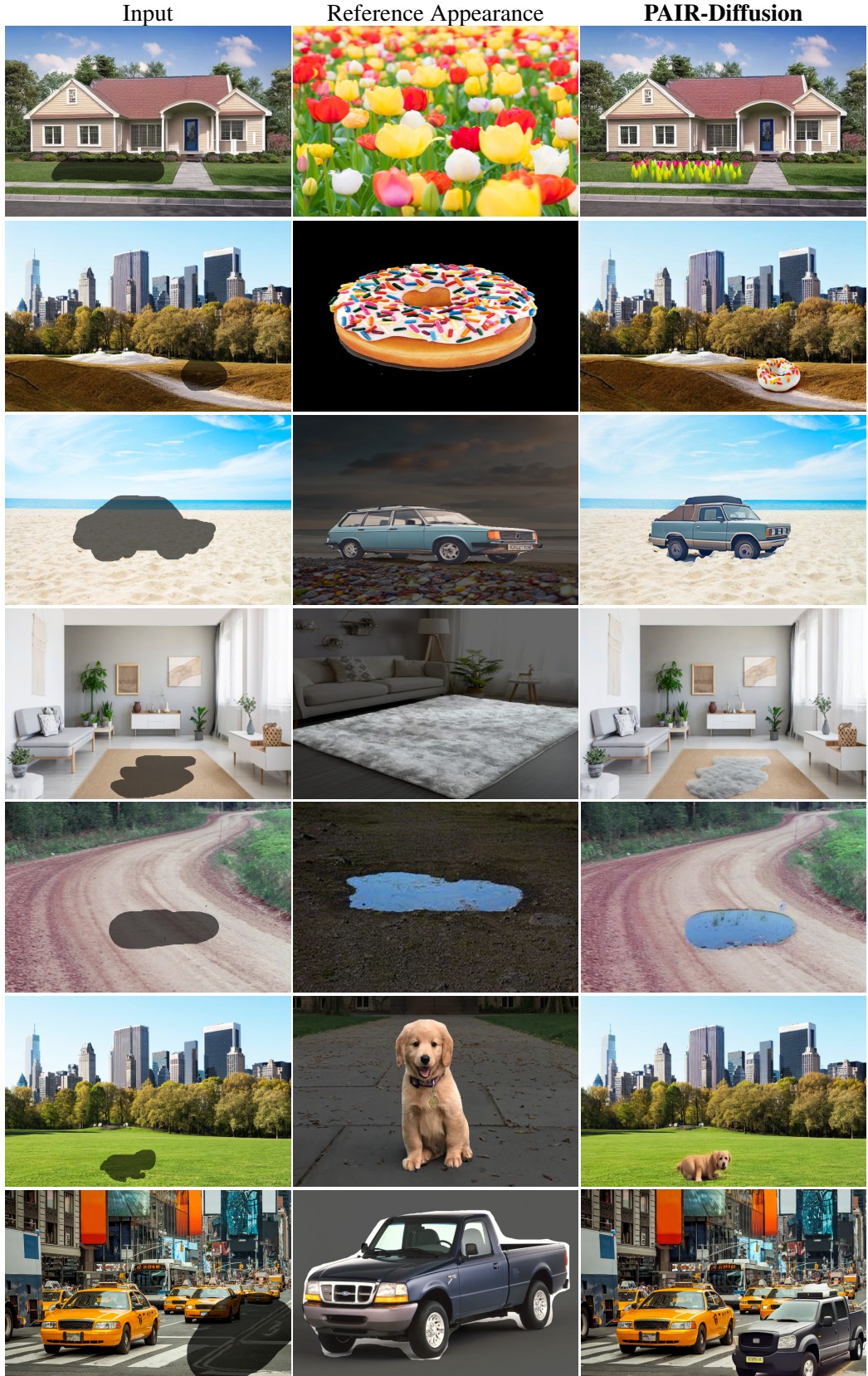

Figure 12: Add objects to the scene.

Input                                        Variations

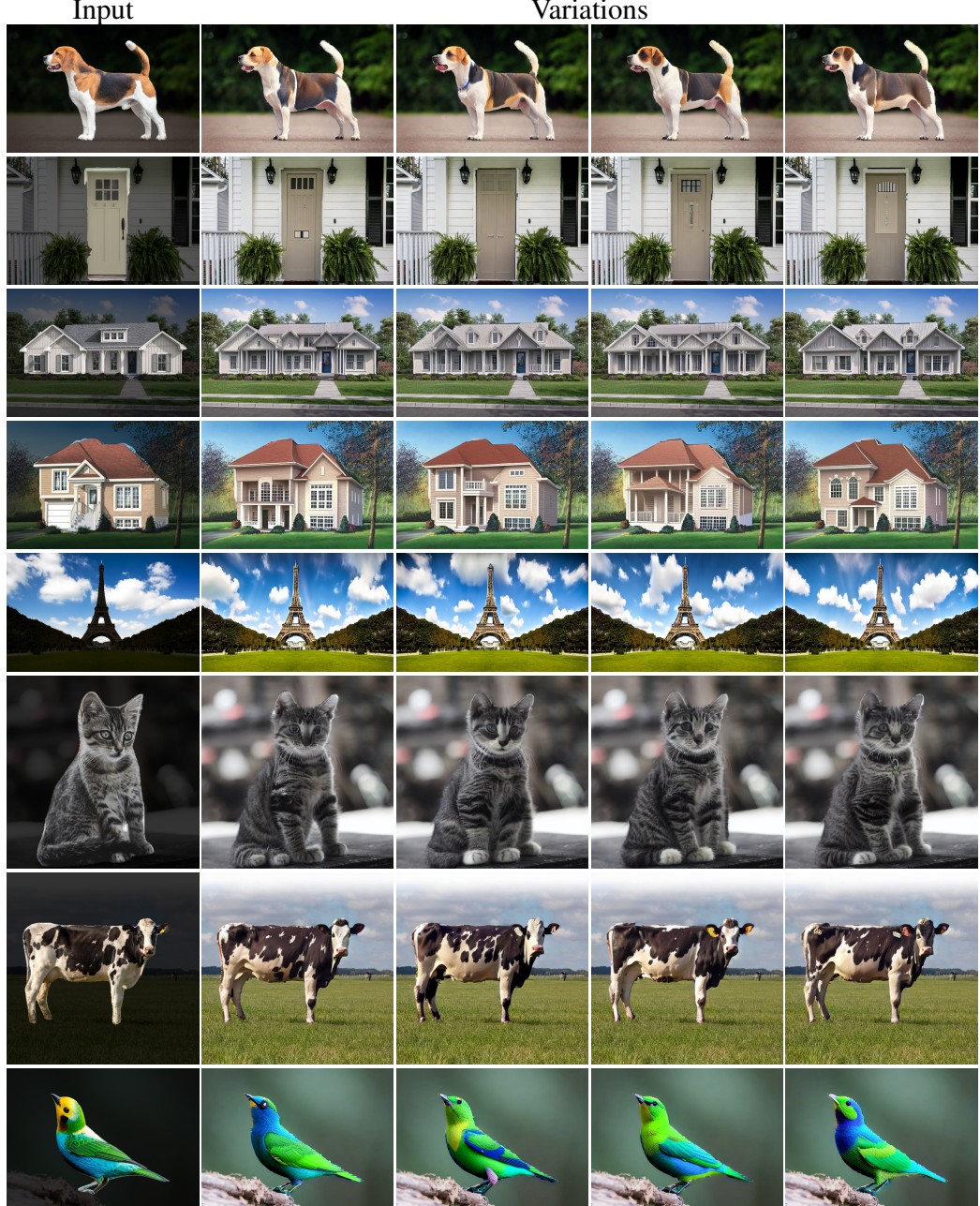

Figure 13: Variations.

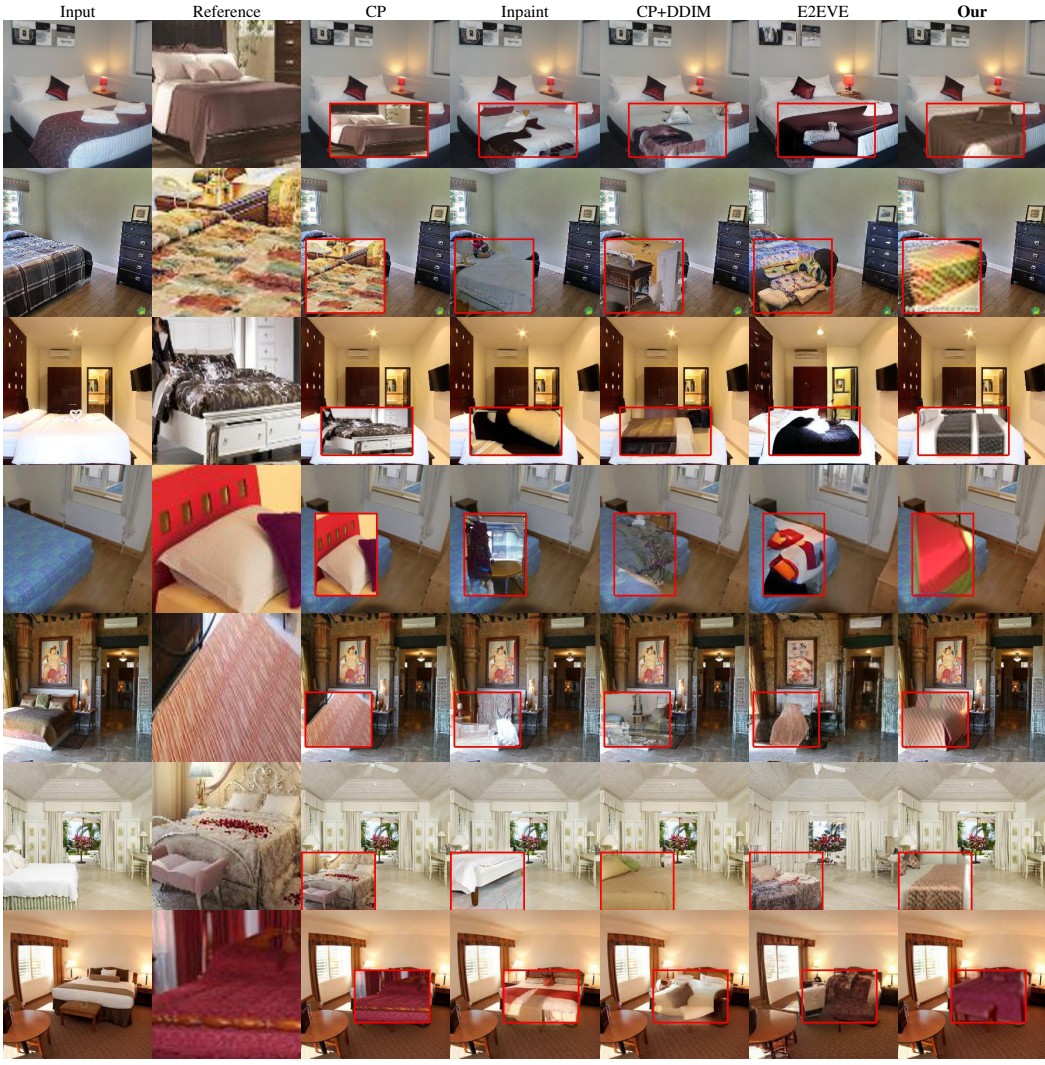

Figure 14: Visual results for in-domain localized image editing. We edit the input image, using as driver the reference image, targeting the red-boxed area. With PAIR-Diffusion we can perform realistic edits in challenging scenarios. For example, in the first row, we can use the entire bed as a driver and edit only a patch of the input image. On the contrary, in the last row, we use a small patch as driver and target the whole bed of the input image for the edit. In both cases, our method outputs realistic edited images. Moreover, due to the masked DDIM technique we introduce almost no distortion in the area outside the red box (*i.e.* the one that should not change). We show results for all the baselines.

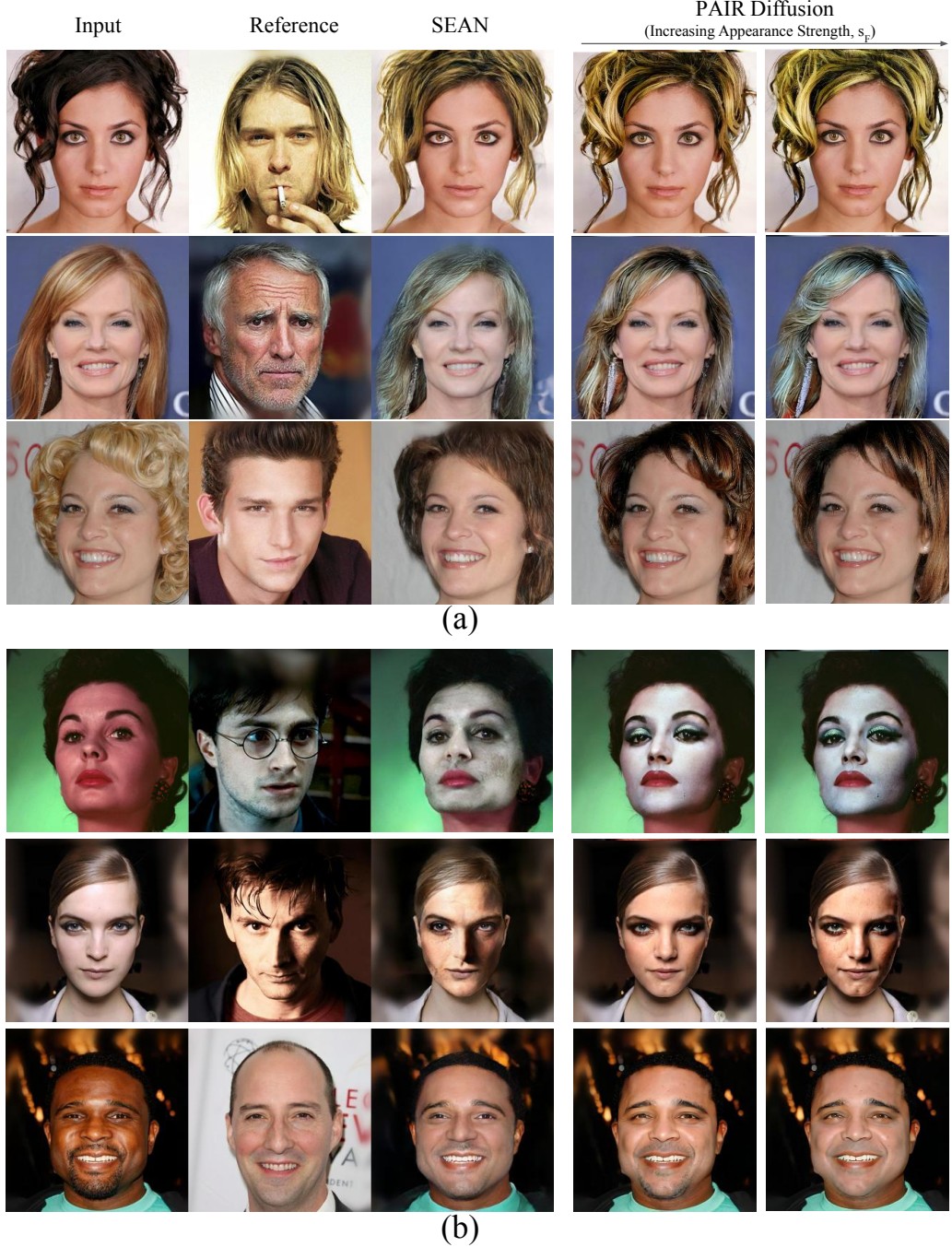

Figure 15: Comparison with SEAN.

Reference Image    Input Image                    Strength of Appearance Manipulation

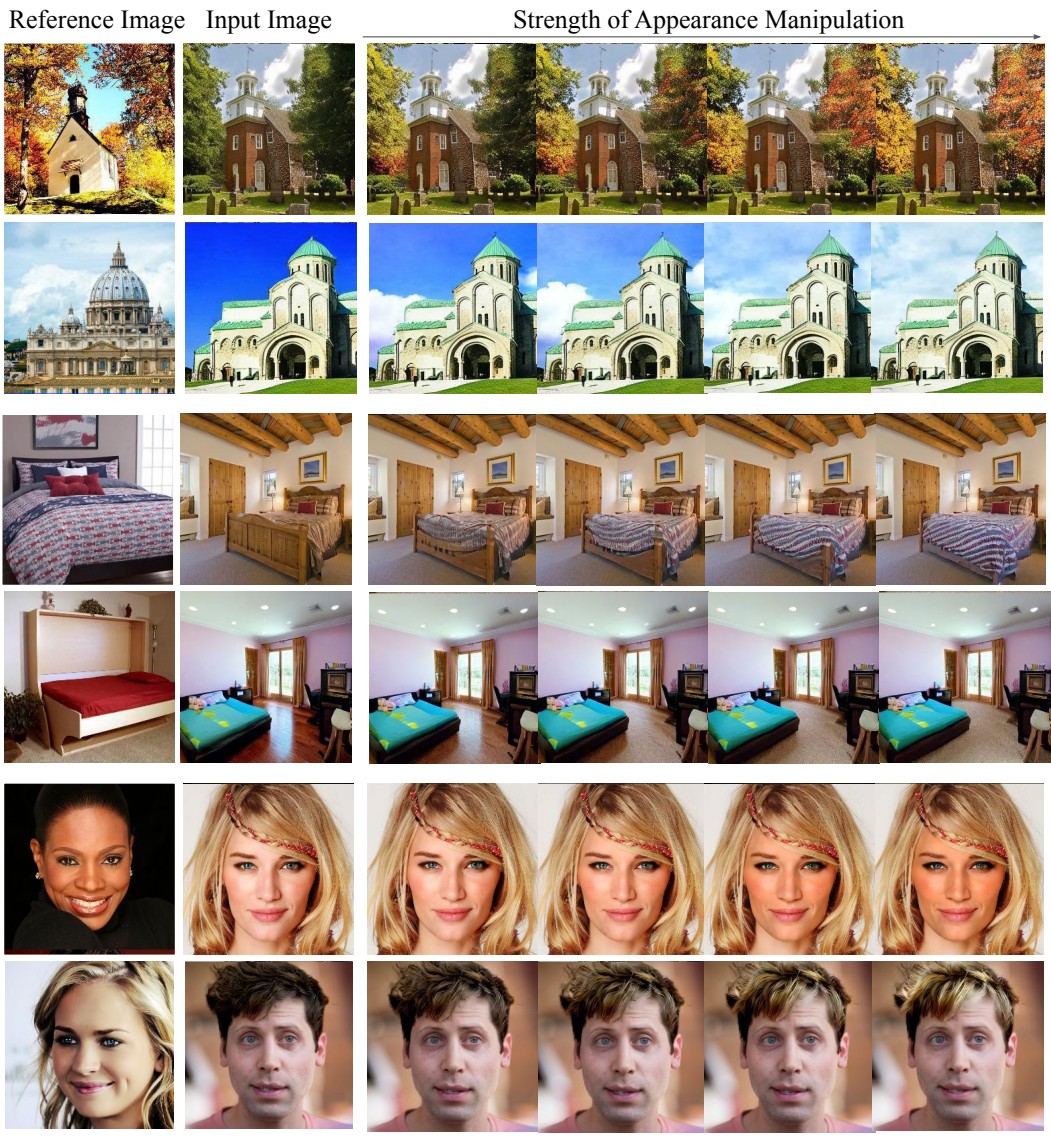

Figure 16: Interpolation results.

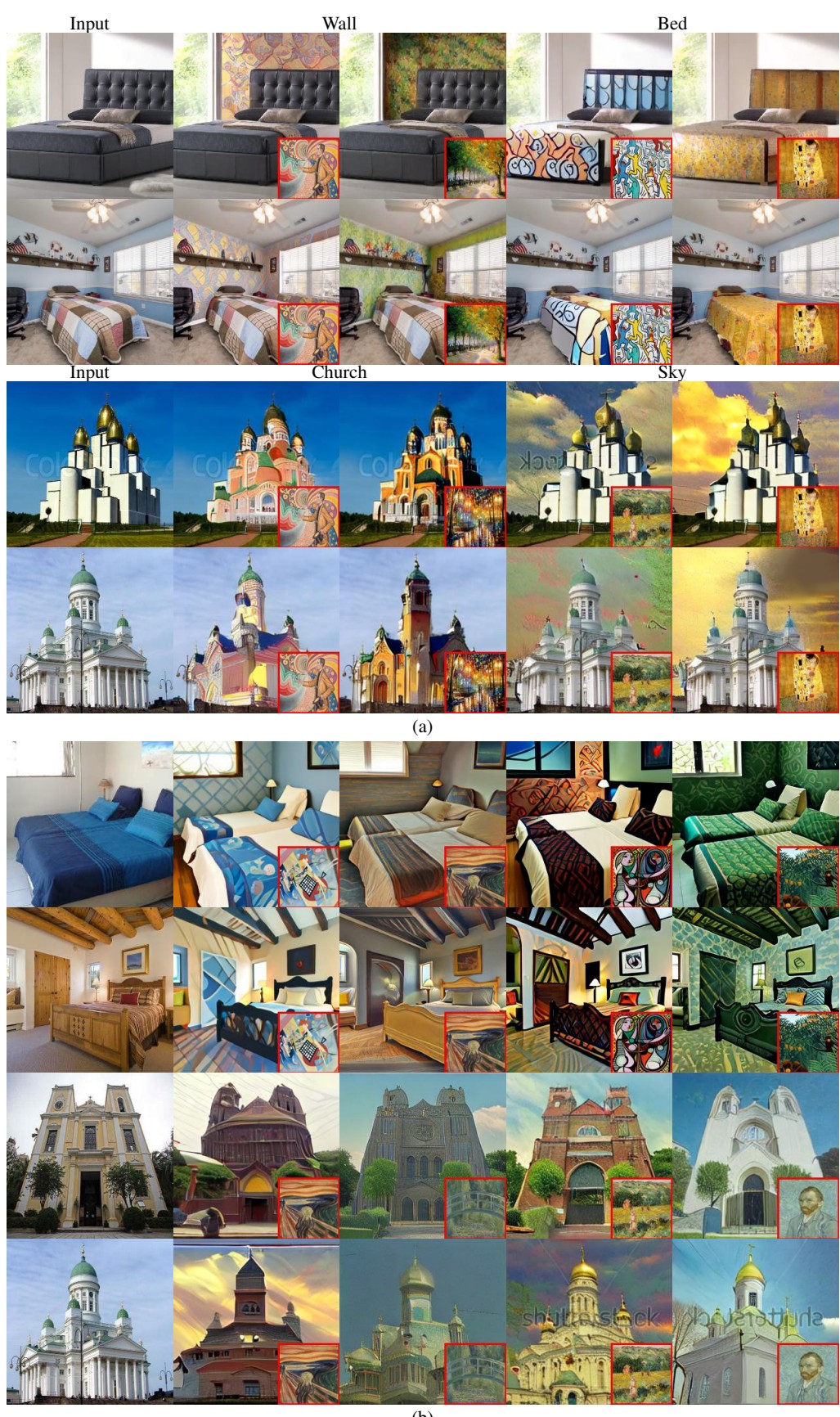

Figure 17: Visual results for (a) local image editing, (b) global image appearance manipulation.