# OpenReview forum: "PAIR Diffusion: A Comprehensive Multimodal Object-Level Image Editor"
_ICLR.cc/2024/Conference — ICLR 2024 Conference Withdrawn Submission_

### Official Review · Reviewer_JFgB · 2023-10-31

**Soundness:** 2 fair
**Presentation:** 3 good
**Contribution:** 3 good
**Rating:** 6
**Confidence:** 3

**Summary:**

The paper introduces the PAIR Diffusion framework, which enables fine-grained control over the properties (appearance and structure) of individual objects in an image, allowing for comprehensive object-level image editing. The framework enables editing operations such as appearance editing, shape editing, adding objects, and variations, without the need for an inversion step. The authors propose multimodal classifier-free guidance, allowing for editing images using both reference images and text. The paper underlines its contributions through qualitative results, demonstrating the framework's effectiveness in enhancing the appearance of objects, whether they are simple or complex.

**Strengths:**

_The paper is well-written and easy to follow.
_The proposed method is captivating and demonstrates its potential by effectively leveraging multiple features per object. This opens up new horizons for fine-grained object editing within this field.
_The authors have also conducted their proof and experiments with diligence and thoroughness, providing robust support for their proposed method.

**Weaknesses:**

_Equation 2 raises a concern regarding the element-wise multiplication between the feature map and the object's shape, given the different sizes of the spatial map (h x w) and the shape (H x W). Moreover, I find the usage of notations j and k in the summation to be ambiguous and would appreciate a more precise definition.
_The paper lacks reported inference times for image encoders, considering the use of CNN and transformer-based methods. This is a notable omission in assessing practicality.
_Addressing issues with object illumination in appearance editing and adding object to scene is necessary, as some results appear less realistic.

**Questions:**

Have you explored the utilization of alternative backbones such as ResNet or EfficientNet for capturing the low-level characteristics of objects?

---

### Official Review · Reviewer_63QG · 2023-10-31

**Soundness:** 3 good
**Presentation:** 3 good
**Contribution:** 2 fair
**Rating:** 6
**Confidence:** 4

**Summary:**

This paper introduces a novel approach to object-level image editing, positing images as composites of distinct objects. The aim is to control the property of each object in a fine-grained manner. Two main attributes are defined for each object structure and appearance. Under the formulation of object-level image editing, the proposed method can handle various image editing tasks. These include localized free-form shape editing, appearance editing, simultaneous shape and appearance editing, addition of objects in a precise manner, and varying the image at the object level.  The authors show promising results on each task. They also propose a classifier-free guidance based inference system to integrate all different modalities.

**Strengths:**

1. **Presentation**: The paper is laudable for its clarity in showcasing its contributions. It provides ample evidence for the method's efficacy through comprehensive evaluations.
2. **User Guidance**: The introduced guidance control setting is intuitive. Users have the flexibility to adjust three parameters, offering them control over the reliance on each aspect of the editing process.

**Weaknesses:**

1. **Discussion on Limitations**: The paper misses out on discussing some potential limitations. For instance, in Figure 3 (first row), even though there's a successful transfer of the general color, the resultant car's identity (structure) deviates from both the source and the reference image. Labeling this as "appearance editing" might be a misrepresentation.

**Questions:**

None

---

### Official Review · Reviewer_GKjx · 2023-11-01

**Soundness:** 3 good
**Presentation:** 3 good
**Contribution:** 3 good
**Rating:** 6
**Confidence:** 4

**Summary:**

PAIR Diffusion framework introduces a novel approach to object-level image editing. By utilizing diffusion models, the authors enable precise control over the structure and appearance of individual objects within images. Their method facilitates a wide range of editing operations, including appearance and shape editing, object addition, and introducing variations. The approach relies on extracting structure information from panoptic segmentation maps and appearance information from VGG and DINOv2 features. These representations are then used to condition a diffusion model, which is trained to generate images. The reviewer would like to highlight the framework's capacity to define various image editing tasks in terms of altering the structure and appearance of individual objects and introduces multimodal, classifier-free guidance, incorporating reference images and text for controlling the output in conjunction with foundational diffusion models.

**Strengths:**

The framework's editing operations are intuitively designed, making it user-friendly and accessible for image editing tasks, ensuring that users can readily manipulate an image's structure and appearance to achieve desired outcomes.

An intriguing and noteworthy aspect of the framework is its ability to perform appearance-level editing at the object level. This unique capability pleasantly surprised the reviewer, highlighting the framework's potential to address specific use cases that were not previously explored in depth.

**Weaknesses:**

It would be valuable to see a detailed comparison of the PAIR Diffusion framework's performance with that of Epstein et al.'s "Diffusion self-guidance for controllable image generation." Such a comparison could shed light on the unique strengths and weaknesses of each approach in the context of image editing and generation.

The use of 2D segmentation maps may introduce geometric distortions in objects, as they lack 3D awareness. Understanding how these limitations impact the framework's ability to accurately represent and edit objects in images is crucial for a comprehensive evaluation of its capabilities and potential improvements.

**Questions:**

Is it possible to achieve identity-preserving edits using the PAIR Diffusion framework, or does it tend to make identities malleable and subject to changes during the editing process? Exploring the framework's capabilities in this regard can offer insights into the preservation of key object or subject attributes.

Understanding how a "dream booth"-like approach functions within the framework would be intriguing. This raises questions about the mechanisms and control it provides in generating personalized content and the degree to which users can influence the creative and imaginative aspects of the editing process.

---

### Official Review · Reviewer_H2Ab · 2023-11-01

**Soundness:** 2 fair
**Presentation:** 2 fair
**Contribution:** 2 fair
**Rating:** 3
**Confidence:** 5

**Summary:**

This paper proposes PAIR-Diffusion, a general framework to enable object-level editing in diffusion models. It allows editing the structure and appearance of each object in the image independently. The proposed design inherently supports various editing tasks using a single model: localized free-form shape editing, appearance editing, editing shape and appearance simultaneously, adding objects in a controlled manner, and object-level image variation.

**Strengths:**

1. It is meaningful to focus on local editing and integrate multiple tasks within a single framework.
2. The proposed method is somewhat reasonable and easy to understand.
3. The paper is overall well-structured.

**Weaknesses:**

1. The designed method performs average pooling to get the appearance feature of local region. How to maintain the local detail features after average pooling? As shown the visualization results, although the overall color and texture are roughly correct, the method does not have the ability to maintain details. For example, in Figure 3 in the main paper, the details of yellow car are changed a lot. In Figure 4 in the main paper, the details of hamburger are changed a lot, although it is still a hamburger. In Figure 12 in the supplementary, the details of car and bagel are also changed a lot. Besides, the designed method is trivial and lacks novelty.
2. Given an input image, without making any change, based on its semantic map and local appearance features, could this input image be exactly constructed? If so, how is this goal achieved? Based on my understanding, compared with the original image, semantic map and local appearance features have lost much detail information. If not, how can the unedited regions remain the same?
3. The main paper does not mention the training details. How many training images are used? How about the generalization ability of the model? Could the model generalize well to the test images which are quite different from the training images?
4. The evaluation details are not very clear. In Section 4.2, is the FID calculated between two images or two image sets? Is L1 distance only calculated within the unedited region? When calculating SSIM between edited region and the target image, is there any guarantee that the edited region and the target image have strict spatial correspondence? Besides, the evaluation metrics are not very comprehensive. For each subtask, only a few baselines are compared, which is not very convincing. The authors should adopt more evaluation metrics and conduct more sufficient comparison.

**Questions:**

1. Provide more explanations for the proposed method and the visualization results.
2. Provide more details of training process and explanations for the model generalization ability.
3. Provide more sufficient experimental justification.

---

### Official Review · Reviewer_BnZE · 2023-11-01

**Soundness:** 3 good
**Presentation:** 3 good
**Contribution:** 1 poor
**Rating:** 3
**Confidence:** 5

**Summary:**

Problem
* Text prompts and low-level conditioning cannot control multiple objects individually.

Goal
* To provide separate control over structure and appearance of individual objects

Method
* The proposed method make the diffusion model to be conditioned on segmentation mask and appearance embedding (pretrained VGG and DINO) following ControlNet.
* Multi-modal (image and text) classifier-free guidance controls strength of each component.

Applications
* Reference-based appearance editing
* Random appearance editing
* Free-form shape editing
* Adding objects

**Strengths:**

1. The proposed method does not require inversion steps.
* Instead, it uses pretrained networks.
2. The proposed method is simple and intuitive.
3. This paper tackles an interesting and useful editing task.

**Weaknesses:**

1. The proposed method is a naive variation of ControlNet: feature maps instead of edge / keypoints / segmentation masks.
2. "We do not require specialized datasets for training" is an overstatement. It requires pretrained segmentation network and pretrained VGG & DINO.
3. The proposed method loses style if the reference image is artistic. e.g., Van Gogh in Figure 1 top center and Figure 17 in supp.
4. Definition of shape editing is vague or it accompanies a wrong experiment.
* Appearance editing edits the appearance of an object *in the input image* (Figure 3).
* Object appearance variation edits the appearance  of an object *in the input image* (Figure 13 in supp).
* On the other hand, shape editing edits the shape of an object *in the reference image* (Figure 4).
6. Quant. evaluation for semantic image synthesis covers only CelebA-HQ. It would be more informative to add CityScapes / ADE20K / Facades as in SEAN.

(minor)

typos
* off the shelf -> off-the-shelf
* we use them to realize the Eq. 1.  -> remove "the"
* ... many more. Running a grammar checker may help.

Please do not use "can" where it has nothing to do with capability. E.g., "We can use the method described in ..." -> We use ...

**Questions:**

1. What happens if the prompt and the reference do not agree?

Please refer to Weaknesses for my concerns toward rejection, especially #1.